# Labeling Trick: A Theory of Using Graph Neural Networks for Multi-Node Representation Learning

**Muhan Zhang**[1,2,∗]  **Pan Li**[3,†]  **Yinglong Xia**[4]   **Kai Wang**[4]   **Long Jin**[4]

[1]Institute for Artificial Intelligence, Peking University
[2]Beijing Institute for General Artificial Intelligence
[3]Department of Computer Science, Purdue University
[4]Facebook AI

## Abstract

In this paper, we provide a theory of using graph neural networks (GNNs) for multi-node representation learning (where we are interested in learning a representation for a set of more than one node, such as link). We know that GNN is designed to learn single-node representations. When we want to learn a node set representation involving multiple nodes, a common practice in previous works is to directly aggregate the single-node representations obtained by a GNN into a joint node set representation. In this paper, we show a fundamental constraint of such an approach, namely the inability to capture the dependence between nodes in the node set, and argue that directly aggregating individual node representations does not lead to an effective joint representation for multiple nodes. Then, we notice that a few previous successful works for multi-node representation learning, including SEAL, Distance Encoding, and ID-GNN, all used node labeling. These methods first label nodes in the graph according to their relationships with the target node set before applying a GNN. Then, the node representations obtained in the labeled graph are aggregated into a node set representation. By investigating their inner mechanisms, we unify these node labeling techniques into a single and most general form—*labeling trick*. We prove that with labeling trick a sufficiently expressive GNN learns the most expressive node set representations, thus in principle solves any joint learning tasks over node sets. Experiments on one important two-node representation learning task, link prediction, verified our theory. Our work explains the superior performance of previous node-labeling-based methods, and establishes a theoretical foundation of using GNNs for multi-node representation learning.

## 1   Introduction

Graph neural networks (GNNs) [1–10] have achieved great successes in recent years. While GNNs have been well studied for single-node tasks (such as node classification) and whole-graph tasks (such as graph classification), using GNNs to predict a set of multiple nodes is less studied and less understood. Among such *multi-node representation learning* problems, *link prediction* (predicting the link existence/class/value between a set of two nodes) is perhaps the most important one due to its wide applications in practice, including friend recommendation in social networks [11], movie recommendation in Netflix [12], protein interaction prediction [13], drug response prediction [14], knowledge graph completion [15], etc. In this paper, we use link prediction as a medium to study GNN's multi-node representation learning ability. Note that although our examples and experiments are all around link prediction, our theory applies generally to all multi-node representation learning problems such as triplet [16], motif [17] and subgraph [18] prediction tasks.

---

[∗]Corresponding author: Muhan Zhang (`muhan@pku.edu.cn`). Work done as a research scientist at Facebook.
[†]Pan Li inspires the study of labeling tricks, proves Theorem 2, and helps check the theoretical framework.

35th Conference on Neural Information Processing Systems (NeurIPS 2021).

There are two main classes of GNN-based link prediction methods: Graph AutoEncoder (GAE) [19] and SEAL [20, 21]. **GAE** (and its variational version VGAE [19]) first applies a GNN to the entire network to compute a representation for each node. The representations of the two end nodes of the link are then aggregated to predict the target link. GAE represents a common practice of using GNNs to learn multi-node representations. That is, first obtaining individual node representations through a GNN as usual, and then aggregating the representations of those nodes of interest as the multi-node representation. On the contrary, **SEAL** applies a GNN to an enclosing subgraph around each link, where nodes in the subgraph are *labeled differently* according to their distances to the two end nodes before applying the GNN. Despite both using GNNs for link prediction, SEAL often shows much better practical performance than GAE. As we will see, the key lies in SEAL's **node labeling** step.

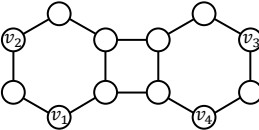

Figure 1: In this graph, nodes $v_2$ and $v_3$ are isomorphic; links $(v_1, v_2)$ and $(v_4, v_3)$ are isomorphic; link $(v_1, v_2)$ and link $(v_1, v_3)$ are **not** isomorphic. However, if we aggregate two node representations learned by a GNN as the link representation, we will give $(v_1, v_2)$ and $(v_1, v_3)$ the same prediction.

We first give a simple example to show when GAE fails. In Figure 1, $v_2$ and $v_3$ have symmetric positions in the graph—from their respective views, they have exactly the **same $h$-hop neighborhood** for any $h$. Thus, without node features, GAE will learn the **same** representation for $v_2$ and $v_3$. However, when we want to predict which one of $v_2$ and $v_3$ is more likely to form a link with $v_1$, GAE will aggregate the representations of $v_1$ and $v_2$ as the link representation of $(v_1, v_2)$, and aggregate the representations of $v_1$ and $v_3$ to represent $(v_1, v_3)$, thus giving $(v_1, v_2)$ and $(v_1, v_3)$ the same representation and prediction. The failure to distinguish links $(v_1, v_2)$ and $(v_1, v_3)$ that have apparently different structural roles in the graph reflects one key limitation of GAE-type methods: by computing $v_1$ and $v_2$'s representations **independently** of each other, GAE cannot capture the **dependence** between two end nodes of a link. For example, $(v_1, v_2)$ has a much shorter path between them than that of $(v_1, v_3)$; and $(v_1, v_2)$ has both nodes in the same hexagon, while $(v_1, v_3)$ does not.

Take common neighbor (CN) [22], one elementary heuristic feature for link prediction, as another example. CN counts the number of common neighbors between two nodes to measure their likelihood of forming a link, which is widely used in social network friend recommendation. CN is the foundation of many other successful heuristics such as Adamic-Adar [11] and Resource Allocation [23], which are also based on neighborhood overlap. However, GAE **cannot** capture such neighborhood-overlap-based features. This can be seen from Figure 1 too. There is 1 common neighbor between $(v_1, v_2)$ and 0 between $(v_1, v_3)$, but GAE always gives $(v_1, v_2)$ and $(v_1, v_3)$ the same representation. The failure to learn common neighbor demonstrates GAE's severe limitation for link prediction. The root cause still lies in that GAE computes node representations independently of each other—when computing the representation of one end node, it is **not aware** of the other end node.

One way to alleviate the above failure is to use one-hot encoding of node indices or random features as input node features [24, 25]. With such node-discriminating features, $v_2$ and $v_3$ will have different node representations, thus $(v_1, v_2)$ and $(v_1, v_3)$ may also have different link representations after aggregation, enabling GAE to discriminate $(v_1, v_2)$ and $(v_1, v_3)$. However, using node-discriminating features loses GNN's **inductive learning** ability to map nodes and links with identical neighborhoods (such as nodes $v_2$ and $v_3$, and links $(v_1, v_2)$ and $(v_4, v_3)$) to the same representation, which results in a great loss of generalization ability. The resulting model is no longer permutation invariant/equivariant, violating the fundamental design principle of GNNs. Is there a way to improve GNNs' link discriminating power (so that links like $(v_1, v_2)$ and $(v_1, v_3)$ can be distinguished), while maintaining their inductive learning ability (so that links $(v_1, v_2)$ and $(v_4, v_3)$ have the same representation)?

In this paper, we analyze the above problem from a *structural representation learning* point of view. Srinivasan and Ribeiro [26] prove that the multi-node prediction problem on graphs ultimately only requires finding a *most expressive structural representation* of node sets, which gives two node sets the same representation if and only if they are *isomorphic* (a.k.a. symmetric, on the same orbit) in the graph. For example, link $(v_1, v_2)$ and link $(v_4, v_3)$ in Figure 1 are isomorphic. A most expressive structural representation for links should give any two isomorphic links the same representation while discriminating all non-isomorphic links (such as $(v_1, v_2)$ and $(v_1, v_3)$). According to our discussion above, GAE-type methods that directly aggregate node representations cannot learn a most expressive structural representation. Then, how to learn a most expressive structural representation of node sets?

To answer this question, we revisit the other GNN-based link prediction framework, SEAL, and analyze how node labeling helps a GNN learn better node set representations. We find out that two properties of a node labeling are crucial for its effectiveness: 1) *target-nodes-distinguishing* and 2) *permutation equivariance*. With these two properties, we define *labeling trick* (Section 4.1), which unifies previous node labeling methods into a single and most general form. Theoretically, we prove that with labeling trick a sufficiently expressive GNN can learn most expressive structural representations of node sets (Theorem 1), which reassures GNN's node set prediction ability. It also closes the gap between GNN's node representation learning nature and node set tasks' multi-node representation learning requirement. We further extend our theory to *local isomorphism* (Section 5). And finally, experiments on four OGB link existence prediction datasets [27] verified our theory.

Note that the labeling trick theory allows the presence of node/edge features/types, thus is not restricted to non-attributed and homogeneous graphs. Previous works on heterogeneous graphs, such as knowledge graphs [28] and recommender systems [29] have already seen successful applications of labeling trick. Labeling trick is also not restricted to two-node link representation learning tasks, but generally applies to any multi-node representation learning tasks.

## 2   Preliminaries

In this section, we introduce some important concepts that will be used in the analysis of the paper, including *permutation*, *set isomorphism* and *most expressive structural representation*.

We consider a graph $\mathcal{G} = (V, E, \mathbf{A})$, where $V = \{1, 2, \ldots, n\}$ is the set of $n$ vertices, $E \subseteq V \times V$ is the set of edges, and $\mathbf{A} \in \mathbb{R}^{n \times n \times k}$ is a 3-dimensional tensor containing node and edge features. The diagonal components $\mathbf{A}_{i,i,:}$ denote features of node $i$, and the off-diagonal components $\mathbf{A}_{i,j,:}$ denote features of edge $(i, j)$. For heterogeneous graphs, the node/edge types can also be expressed in $\mathbf{A}$ using integers or one-hot encoding vectors. We further use $A \in \{0, 1\}^{n \times n}$ to denote the adjacency matrix of $\mathcal{G}$ with $A_{i,j} = 1$ iff $(i, j) \in E$. We let $A$ be the first slice of $\mathbf{A}$, i.e., $A = \mathbf{A}_{:,:,1}$. Since $\mathbf{A}$ contains the complete information of a graph, we sometimes directly use $\mathbf{A}$ to denote the graph.

**Definition 1.** *A **permutation** $\pi$ is a bijective mapping from $\{1, 2, \ldots, n\}$ to $\{1, 2, \ldots, n\}$. Depending on the context, $\pi(i)$ can mean assigning a new index to node $i \in V$, or mapping node $i$ to node $\pi(i)$ of another graph. All $n!$ possible $\pi$'s constitute the permutation group $\Pi_n$. For joint prediction tasks over a set of nodes, we use $S$ to denote the **target node set**. For example, $S = \{i, j\}$ if we want to predict the link between $i, j$. We define $\pi(S) = \{\pi(i) | i \in S\}$. We further define the permutation of $\mathbf{A}$ as $\pi(\mathbf{A})$, where $\pi(\mathbf{A})_{\pi(i),\pi(j),:} = \mathbf{A}_{i,j,:}$.*

Next, we define *set isomorphism*, which generalizes graph isomorphism to arbitrary node sets.

**Definition 2.** *(Set isomorphism) Given two $n$-node graphs $\mathcal{G} = (V, E, \mathbf{A})$, $\mathcal{G}' = (V', E', \mathbf{A}')$, and two node sets $S \subseteq V$, $S' \subseteq V'$, we say $(S, \mathbf{A})$ and $(S', \mathbf{A}')$ are isomorphic (denoted by $(S, \mathbf{A}) \simeq (S', \mathbf{A}')$) if $\exists \pi \in \Pi_n$ such that $S = \pi(S')$ and $\mathbf{A} = \pi(\mathbf{A}')$.*

When $(V, \mathbf{A}) \simeq (V', \mathbf{A}')$, we say two graphs $\mathcal{G}$ and $\mathcal{G}'$ are *isomorphic* (abbreviated as $\mathbf{A} \simeq \mathbf{A}'$ because $V = \pi(V')$ for any $\pi$). Note that set isomorphism is **more strict** than graph isomorphism, because it not only requires graph isomorphism, but also requires the permutation maps a specific node set $S$ to another node set $S'$. In practice, when $S \neq V$, we are often more concerned with the case of $\mathbf{A} = \mathbf{A}'$, where isomorphic node sets are defined **in the same graph** (automorphism). For example, when $S = \{i\}$, $S' = \{j\}$ and $(i, \mathbf{A}) \simeq (j, \mathbf{A})$, we say nodes $i$ and $j$ are isomorphic in graph $\mathbf{A}$ (or they have symmetric positions/same structural role in graph $\mathbf{A}$). An example is $v_2$ and $v_3$ in Figure 1.

We say a function $f$ defined over the space of $(S, \mathbf{A})$ is *permutation invariant* (or *invariant* for abbreviation) if $\forall \pi \in \Pi_n$, $f(S, \mathbf{A}) = f(\pi(S), \pi(\mathbf{A}))$. Similarly, $f$ is *permutation equivariant* if $\forall \pi \in \Pi_n$, $\pi(f(S, \mathbf{A})) = f(\pi(S), \pi(\mathbf{A}))$. Permutation invariance/equivariance ensures representations learned by a GNN is invariant to node indexing, which is a fundamental design principle of GNNs.

Now we define most expressive structural representation of a node set, following [26, 21]. Basically, it assigns a unique representation to each equivalence class of isomorphic node sets.

**Definition 3.** *Given an invariant function $\Gamma(\cdot)$, $\Gamma(S, \mathbf{A})$ is a **most expressive structural representation** for $(S, \mathbf{A})$ if $\forall S, \mathbf{A}, S', \mathbf{A}'$, $\Gamma(S, \mathbf{A}) = \Gamma(S', \mathbf{A}') \Leftrightarrow (S, \mathbf{A}) \simeq (S', \mathbf{A}')$.*

For simplicity, we will briefly use *structural representation* to denote most expressive structural representation in the rest of the paper. We will omit **A** if it is clear from context. We call $\Gamma(i, \mathbf{A})$ a *structural node representation* for $i$, and call $\Gamma(\{i, j\}, \mathbf{A})$ a *structural link representation* for $(i, j)$.

Definition 3 requires that the structural representations of two node sets are the same if and only if the two node sets are isomorphic. That is, isomorphic node sets always have the **same** structural representation, while non-isomorphic node sets always have **different** structural representations. This is in contrast to *positional node embeddings* such as DeepWalk [30] and matrix factorization [31], where two isomorphic nodes can have different node embeddings [32]. GAE using node-discriminating features also learns positional node embeddings.

**Why do we study structural representations?** Formally speaking, Srinivasan and Ribeiro [26] prove that any joint prediction task over node sets only requires a structural representation of node sets. They show that positional node embeddings carry no more information beyond that of structural representations. Intuitively speaking, it is because two isomorphic nodes in a network are perfectly symmetric and interchangeable with each other, and should be indistinguishable from any perspective. Learning a structural node representation guarantees that isomorphic nodes are always classified into the same class. Similarly, learning a structural link representation guarantees isomorphic links, such as $(v_1, v_2)$ and $(v_4, v_3)$ in Figure 1, are always predicted the same, while non-isomorphic links, such as $(v_1, v_2)$ and $(v_1, v_3)$, are always distinguishable, which is not guaranteed by positional node embeddings. Structural representation characterizes the maximum representation power a model can reach on graphs. We use it to study GNNs' multi-node representation learning ability.

## 3 The limitation of directly aggregating node representations

In this section, using GAE for link prediction as an example, we show the key limitation of directly aggregating node representations as a node set representation.

### 3.1 GAE for link prediction

Given a graph **A**, GAE methods [19] first use a GNN to compute a node representation $z_i$ for each node $i$, and then use an aggregation function $f(\{z_i, z_j\})$ to predict link $(i, j)$:

$$\hat{\boldsymbol{A}}_{i,j} = f(\{\boldsymbol{z}_i, \boldsymbol{z}_j\}), \text{ where } \boldsymbol{z}_i = \text{GNN}(i, \mathbf{A}), \boldsymbol{z}_j = \text{GNN}(j, \mathbf{A}).$$

Here $\hat{\boldsymbol{A}}_{i,j}$ is the predicted score for link $(i, j)$. The model is trained to maximize the likelihood of reconstructing the true adjacency matrix. The original GAE uses a two-layer GCN [5] as the GNN, and let $f(\{\boldsymbol{z}_i, \boldsymbol{z}_j\}) := \sigma(\boldsymbol{z}_i^\top \boldsymbol{z}_j)$. In principle, we can replace GCN with any GNN, and replace $\sigma(\boldsymbol{z}_i^\top \boldsymbol{z}_j)$ with an MLP over any aggregation function over $\{\boldsymbol{z}_i, \boldsymbol{z}_j\}$. Besides inner product, other aggregation choices include mean, sum, bilinear product, concatenation, and Hadamard product. In the following, we will use GAE to denote a general class of GNN-based link prediction methods.

GAE uses a GNN to learn node representations and then aggregates pairwise node representations as link representations. Two natural questions to ask are: 1) Is the node representation learned by the GNN a *structural node representation*? 2) Is the link representation aggregated from two node representations a *structural link representation*? We answer them respectively in the following.

### 3.2 GNN and structural node representation

Practical GNNs [33] usually simulate the 1-dimensional Weisfeiler-Lehman (1-WL) test [34] to iteratively update each node's representation by aggregating its neighbors' representations. We use *1-WL-GNN* to denote a GNN with 1-WL discriminating power, such as GIN [35].

A 1-WL-GNN ensures that isomorphic nodes always have the same representation. But the opposite direction is not guaranteed. For example, a 1-WL-GNN gives the same representation to all nodes in an $r$-regular graph. Despite this, 1-WL is known to discriminate almost all non-isomorphic nodes [36]. This indicates that a 1-WL-GNN can always give the same representation to isomorphic nodes, and can give different representations to **almost all** non-isomorphic nodes.

To study GNN's maximum expressive power for multi-node representation learning, we also define a *node-most-expressive GNN*, which gives different representations to **all** non-isomorphic nodes.

**Definition 4.** *A GNN is **node-most-expressive** if* $\forall i, \mathbf{A}, j, \mathbf{A}'$, $\mathrm{GNN}(i, \mathbf{A}) = \mathrm{GNN}(j, \mathbf{A}') \Leftrightarrow (i, \mathbf{A}) \simeq (j, \mathbf{A}')$.

That is, node-most-expressive GNN learns *structural node representations*[3]. We define such a GNN because we want to answer: whether GAE, even equipped with a node-most-expressive GNN (so that GNN's node representation power is not a bottleneck), can learn structural link representations.

### 3.3 GAE cannot learn structural link representations

Suppose GAE is equipped with a node-most-expressive GNN which outputs structural node representations. Then the question becomes: does the aggregation of structural node representations of $i$ and $j$ result in a structural *link* representation of $(i, j)$? The answer is no, as shown in previous works [26, 29]. We have also illustrated it in the introduction: In Figure 1, we have two isomorphic nodes $v_2$ and $v_3$, thus $v_2$ and $v_3$ will have the same structural node representation. By aggregating structural node representations, GAE will give $(v_1, v_2)$ and $(v_1, v_3)$ the same link representation. However, $(v_1, v_2)$ and $(v_1, v_3)$ are not isomorphic in the graph. This indicates:

**Proposition 1.** *GAE **cannot** learn structural link representations no matter how expressive node representations a GNN can learn.*

Similarly, we can give examples like Figure 1 for multi-node representation learning problems involving more than two nodes to show that directly aggregating node representations from a GNN does not lead to a structural representation for node sets. The root cause of this problem is that GNN computes node representations independently, without being aware of the other nodes in the target node set $S$. Thus, even GNN learns the most expressive single-node representations, there is never a guarantee that their aggregation is a structural representation of a node set. In other words, the multi-node representation learning problem is **not breakable** into multiple **independent** single-node representation learning problems. If we have to break it, the multiple single-node representation learning problems should be **dependent** on each other.

## 4  Labeling trick for multi-node representation learning

In this section, we first define the general form of *labeling trick*, and use a specific implementation, zero-one labeling trick, to intuitively explain why labeling trick helps GNNs learn better link representations. Next, we present our main theorem showing that labeling trick enables a node-most-expressive GNN to learn structural representations of node sets, which formally characterizes GNN's maximum multi-node representation learning ability. Then, we review SEAL and show it exactly uses one labeling trick. Finally, we discuss other labeling trick implementations in previous works.

### 4.1  Labeling trick

**Definition 5.** *(**Labeling trick**) Given $(S, \mathbf{A})$, we stack a labeling tensor $\mathbf{L}^{(S)} \in \mathbb{R}^{n \times n \times d}$ in the third dimension of $\mathbf{A}$ to get a new $\mathbf{A}^{(S)} \in \mathbb{R}^{n \times n \times (k+d)}$, where $\mathbf{L}$ satisfies: $\forall S, \mathbf{A}, S', \mathbf{A}', \pi \in \Pi_n$,*
*1. (target-nodes-distinguishing) $\mathbf{L}^{(S)} = \pi(\mathbf{L}^{(S')}) \Rightarrow S = \pi(S')$, and*
*2. (permutation equivariance) $S = \pi(S'), \mathbf{A} = \pi(\mathbf{A}') \Rightarrow \mathbf{L}^{(S)} = \pi(\mathbf{L}^{(S')})$.*

To explain a bit, labeling trick assigns a label vector to each node/edge in graph $\mathbf{A}$, which constitutes the labeling tensor $\mathbf{L}^{(S)}$. By concatenating $\mathbf{A}$ and $\mathbf{L}^{(S)}$, we get the new labeled graph $\mathbf{A}^{(S)}$. By definition we can assign labels to both nodes and edges. However, in this paper, we **only consider node labels** for simplicity, i.e., we let the off-diagonal components $\mathbf{L}^{(S)}_{i,j,:}$ be all zero.

The labeling tensor $\mathbf{L}^{(S)}$ should satisfy two properties in Definition 5. Property 1 requires that if a permutation $\pi$ preserving node labels (i.e., $\mathbf{L}^{(S)} = \pi(\mathbf{L}^{(S')})$) exists between nodes of $\mathbf{A}$ and $\mathbf{A}'$, then the nodes in $S'$ must be mapped to nodes in $S$ by $\pi$ (i.e., $S = \pi(S')$). A sufficient condition for property 1 is to make the target nodes $S$ have *distinct labels* from those of the rest nodes, so that $S$ is

---

[3]Although a polynomial-time implementation is not known for node-most-expressive GNNs, many practical softwares can discriminate all non-isomorphic nodes quite efficiently [37], which provides a promising direction.

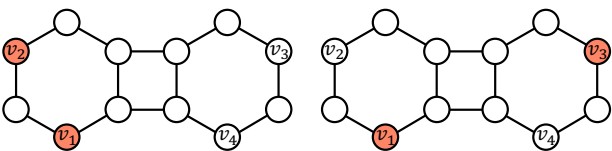

Figure 2: When we predict $(v_1, v_2)$, we will label these two nodes differently from the rest, so that a GNN is aware of the target link when learning $v_1$ and $v_2$'s representations. Similarly, when predicting $(v_1, v_3)$, nodes $v_1$ and $v_3$ will be labeled differently. This way, the representation of $v_2$ in the left graph will be different from that of $v_3$ in the right graph, enabling GNNs to distinguish the non-isomorphic links $(v_1, v_2)$ and $(v_1, v_3)$.

*distinguishable* from others. Property 2 requires that when $(S, \mathbf{A})$ and $(S', \mathbf{A}')$ are isomorphic under $\pi$ (i.e., $S = \pi(S'), \mathbf{A} = \pi(\mathbf{A}')$), the corresponding nodes $i \in S, j \in S', i = \pi(j)$ must always have the same label (i.e., $\mathbf{L}^{(S)} = \pi(\mathbf{L}^{(S')})$). A sufficient condition for property 2 is to make the labeling function *permutation equivariant*, i.e., when the target $(S, \mathbf{A})$ changes to $(\pi(S), \pi(\mathbf{A}))$, the labeling tensor $\mathbf{L}^{(S)}$ should equivariantly change to $\pi(\mathbf{L}^{(S)})$.

Now we introduce a simplest labeling trick satisfying the two properties in Definition 5, and use it to illustrate how labeling trick helps GNNs learn better node set representations.

**Definition 6.** *(Zero-one labeling trick) Given a graph $\mathbf{A}$ and a set of nodes $S$ to predict, we give it a diagonal labeling matrix $\mathbf{L}^{(S)} \in \mathbb{R}^{n \times n \times 1}$ such that $\mathbf{L}^{(S)}_{i,i,1} = 1$ if $i \in S$ and $\mathbf{L}^{(S)}_{i,i,1} = 0$ otherwise.*

In other words, the zero-one labeling trick assigns label 1 to nodes in $S$, and label 0 to all nodes not in $S$. It is a valid labeling trick because firstly, nodes in $S$ get *distinct labels*, and secondly, the labeling function is *permutation equivariant* by always giving nodes in the target node set a label 1. These node labels serve as additional node features fed to a GNN together with the original node features.

Let's return to the example in Figure 1 to see how the zero-one labeling trick helps GNNs learn better link representations. This time, when we want to predict link $(v_1, v_2)$, we will label $v_1, v_2$ differently from the rest nodes, as shown by the different color in Figure 2 left. With nodes $v_1$ and $v_2$ labeled, when the GNN is computing $v_2$'s representation, it is also "aware" of the source node $v_1$, instead of the previous agnostic way that treats $v_1$ the same as other nodes. Similarly, when we want to predict link $(v_1, v_3)$, we will again label $v_1, v_3$ differently from other nodes as shown in Figure 2 right. This way, $v_2$ and $v_3$'s node representations are no longer the same in the two differently labeled graphs (due to the presence of the labeled $v_1$), and we are able to predict $(v_1, v_2)$ and $(v_1, v_3)$ differently. The key difference from GAE is that the node representations are no longer computed independently, but are *conditioned* on each other in order to capture the dependence between nodes.

At the same time, isomorphic links, such as $(v_1, v_2)$ and $(v_4, v_3)$, will still have the same representation, since the zero-one labeled graph for $(v_1, v_2)$ is still symmetric to the zero-one labeled graph for $(v_4, v_3)$. This brings an exclusive advantage over GAE using node-discriminating features.

With $v_1$ and $v_2$ labeled, a GNN can also learn their common neighbor easily: in the first iteration, only $(v_1, v_2)$'s common neighbors will receive the distinct message from both $v_1$ and $v_2$; then in the next iteration, all common neighbors will pass their distinct messages back to both $v_1$ and $v_2$, which effectively encode the number of common neighbors into $v_1$ and $v_2$'s updated representations.

Now we introduce our main theorem showing that with a valid labeling trick, a node-most-expressive GNN can *learn structural representations of node sets*.

**Theorem 1.** *Given a node-most-expressive GNN and an injective set aggregation function AGG, for any $S, \mathbf{A}, S', \mathbf{A}'$, GNN$(S, \mathbf{A}^{(S)}) = $ GNN$(S', \mathbf{A}'^{(S')}) \Leftrightarrow (S, \mathbf{A}) \simeq (S', \mathbf{A}')$, where GNN$(S, \mathbf{A}^{(S)}) :=$ AGG$(\{\text{GNN}(i, \mathbf{A}^{(S)}) | i \in S\})$.*

We include all proofs in the appendix. Theorem 1 implies that AGG$(\{\text{GNN}(i, \mathbf{A}^{(S)}) | i \in S\})$ is a structural representation for $(S, \mathbf{A})$. Remember that directly aggregating the structural node representations learned from the original graph $\mathbf{A}$ does not lead to structural representations of node sets (Section 3.3). Theorem 1 shows that aggregating the structural node representations learned from the **labeled** graph $\mathbf{A}^{(S)}$, somewhat surprisingly, results in a structural representation for $(S, \mathbf{A})$.

The significance of Theorem 1 is that it closes the gap between GNN's single-node representation nature and node set prediction problems' multi-node representation requirement. It demonstrates that

GNNs are able to learn most expressive structural representations of node sets, thus are suitable for joint prediction tasks over node sets too. This answers the open question raised in [26] questioning GNNs' link prediction ability: *are structural node representations in general–and GNNs in particular–fundamentally incapable of performing link (dyadic) and multi-ary (polyadic) prediction tasks?* With Theorem 1, we argue the answer is no. Although GNNs alone have severe limitations for learning joint representations of multiple nodes, GNNs + labeling trick can learn structural representations of node sets too by aggregating structural node representations obtained in the labeled graph.

Theorem 1 assumes a node-most-expressive GNN. To augment Theorem 1, we give the following theorem, which demonstrates labeling trick's power for 1-WL-GNNs.

**Theorem 2.** *In any non-attributed graph with $n$ nodes, if the degree of each node in the graph is between 1 and $\mathcal{O}(\log^{\frac{1-\epsilon}{2h}} n)$ for any constant $\epsilon > 0$, then there exists $\omega(n^{2\epsilon})$ many pairs of non-isomorphic links $(u, w), (v, w)$ such that an $h$-layer 1-WL-GNN gives $u, v$ the same representation, while with labeling trick the 1-WL-GNN gives $u, v$ different representations.*

Theorem 2 shows that in any non-attributed graph there exists a large number ($\omega(n^{2\epsilon})$) of link pairs (like the examples $(v_1, v_2)$ and $(v_1, v_3)$ in Figure 1) which are not distinguishable by 1-WL-GNNs alone but distinguishable by 1-WL-GNNs + labeling trick.

## 4.2 SEAL uses a labeling trick

SEAL [20] is a state-of-the-art link prediction method based on GNNs. It first extracts an *enclosing subgraph* ($h$-hop subgraph) around each target link to predict.

**Definition 7.** *(Enclosing subgraph) Given $(S, \mathbf{A})$, the $h$-hop enclosing subgraph $\mathbf{A}_{(S,h)}$ of $S$ is the subgraph induced from $\mathbf{A}$ by $\cup_{j \in S}\{i \mid d(i, j) \leq h\}$, where $d(i, j)$ is the shortest path distance between nodes $i$ and $j$.*

Then, SEAL applies Double Radius Node Labeling (DRNL) to give an integer label to each node in the enclosing subgraph. DRNL assigns different labels to nodes with **different distances** to the two end nodes of the link. It works as follows: The two end nodes are always labeled 1. Nodes farther away from the two end nodes get larger labels (starting from 2). For example, nodes with distances $\{1, 1\}$ to the two end nodes will get label 2, and nodes with distances $\{1, 2\}$ to the two end nodes will get label 3. So on and so forth. Finally the labeled enclosing subgraph is fed to a GNN to learn the link representation and output the probability of link existence.

**Theorem 3.** *DRNL is a labeling trick.*

Theorem 3 is easily proved by noticing: across different subgraphs, 1) nodes with label 1 are always those in the target node set $S$, and 2) nodes with the same distances to $S$ always have the same label, while distances are permutation equivariant. Thus, SEAL exactly uses a specific labeling trick to enhance its power, which explains its often superior performance than GAE [20].

SEAL only uses a subgraph $\mathbf{A}_{(S,h)}$ within $h$ hops from the target link instead of using the whole graph. This is not a constraint but rather a practical consideration (just like GAE typically uses less than 3 message passing layers in practice), and its benefits will be discussed in detail in Section 5. When $h \to \infty$, the subgraph becomes the entire graph, and SEAL is able to *learn structural link representations* from the labeled (entire) graph.

**Proposition 2.** *When $h \to \infty$, SEAL can learn structural link representations with a node-most-expressive GNN.*

## 4.3 Discussion

**DE and DRNL** In [21], SEAL's distance-based node labeling scheme is generalized to *Distance Encoding* (DE) that can be applied to $|S| > 2$ problems. Basically, DRNL is equivalent to DE-2 using shortest path distance. Instead of encoding two distances into one integer label, DE injectively aggregates the embeddings of two distances into a label vector. **DE is also a valid labeling trick**, as it can also distinguish $S$ and is permutation equivariant. However, there are some subtle differences between DE and DRNL's implementations, which are discussed in Appendix D.

**ID-GNN** You et al. [38] propose *Identity-aware GNN* (ID-GNN), which assigns a unique color to the "identity" nodes and performs message passing for them with a different set of parameters.

ID-GNN's coloring scheme is similar to the zero-one labeling trick that distinguishes nodes in the target set with 0/1 labels. However, when used for link prediction, ID-GNN only colors the source node, while the zero-one labeling trick labels both the source and destination nodes. Thus, ID-GNN can be seen as using a partial labeling trick. The idea of conditioning on only the source node is also used in NBFNet [39]. We leave the exploration of partial labeling trick's power for future work.

**Labeling trick for heterogeneous graphs** Since our graph definition **A** allows the presence of node/edge types, our theory applies to heterogeneous graphs, too. In fact, previous works have already successfully used labeling trick for heterogeneous graphs. IGMC [29] uses labeling trick to predict ratings between users and items (recommender systems), where a user node $k$-hop away from the target link receives a label $2k$, and an item node $k$-hop away from the target link receives a label $2k + 1$. It is a valid labeling trick since the target user and item always receive distinct labels 0 and 1. On the other hand, GRAIL [28] applies the DRNL labeling trick to knowledge graph completion.

**Directed case.** Despite that we do not restrict our graphs to be undirected, our node set definition (Definition 2) does not consider the order of nodes in the set (i.e., direction of link when $|S| = 2$). The ordered case assumes $S = (1, 2, 3)$ is different from $S' = (3, 2, 1)$. One way to solve this is to define labeling trick respecting the order of $S$. In fact, if we define $\pi(S) = \big(\pi(S[i]) \mid i = 1, 2, \ldots, |S|\big)$ (where $S[i]$ denotes the $i^{\text{th}}$ element in the ordered set $S$) in Definition 1, and modify our definition of labeling trick using this new definition of permutation, then Theorem 1 still holds.

**Complexity.** Despite the power, labeling trick may introduce extra computational complexity. The reason is that for every node set $S$ to predict, we need to relabel the graph **A** according to $S$ and compute a new set of node representations within the labeled graph. In contrast, GAE-type methods compute node representations only in the original graph. For small graphs, GAE-type methods can compute all node representations first and then predict multiple node sets at the same time, which saves a significant amount of time. However, for large graphs that cannot fit into the GPU memory, mini-batch training (which extracts a neighborhood subgraph for every node set to predict) has to be used for both GAE-type methods and labeling trick, resulting in similar computation cost.

## 5 Local isomorphism: a more practical view of isomorphism

The concept of most expressive structural representation is based on assigning node sets the same representation if and only if they are *isomorphic* to each other in the graph. However, exact isomorphism is not very common. For example, Babai and Kucera [36] prove that at least $(n - \log n)$ nodes in almost all $n$-node graphs are *non-isomorphic* to each other. In practice, 1-WL-GNN also takes up to $\mathcal{O}(n)$ message passing layers to reach its maximum power for discriminating non-isomorphic nodes, making it very hard to really target on finding exactly isomorphic nodes/links.

**Lemma 1.** *Given a graph with $n$ nodes, a 1-WL-GNN takes up to $\mathcal{O}(n)$ message passing layers to discriminate all the nodes that 1-WL can discriminate.*

In this regard, we propose a more practical concept, called *local isomorphism*.

**Definition 8.** *(Local $h$-isomorphism)* $\forall S, \mathbf{A}, S', \mathbf{A}'$, *we say* $(S, \mathbf{A})$ *and* $(S', \mathbf{A}')$ *are locally $h$-isomorphic to each other if* $(S, \mathbf{A}_{(S,h)}) \simeq (S', \mathbf{A}'_{(S',h)})$.

Local $h$-isomorphism only requires the $h$-hop enclosing subgraphs around $S$ and $S'$ are isomorphic, instead of the entire graphs. We argue that this is a more useful definition than isomorphism, because: 1) Exact isomorphism is rare in real-world graphs. 2) Algorithms targeting on exact isomorphism are more likely to overfit. Only assigning the same representations to exactly isomorphic nodes/links may fail to identify a large amount of nodes/links that are not isomorphic but have very similar neighborhoods. Instead, nodes/links *locally isomorphic* to each other may better indicate that they should have the same representation. With local $h$-isomorphism, *all our previous conclusions based on standard isomorphism still apply*. For example, GAE (without node-discriminating features) still cannot discriminate locally $h$-non-isomorphic links. And a node-most-expressive GNN with labeling trick can learn the most expressive structural representations of node sets w.r.t. local $h$-isomorphism, i.e., learn the same representation for two node sets if and only if they are locally $h$-isomorphic:

**Corollary 1.** *Given a node-most-expressive* GNN *and an injective set aggregation function* AGG, *then for any* $S, \mathbf{A}, S', \mathbf{A}', h$, $\text{GNN}(S, \mathbf{A}_{(S,h)}^{(S)}) = \text{GNN}(S', \mathbf{A}_{(S',h)}'^{(S')}) \Leftrightarrow (S, \mathbf{A}_{(S,h)}) \simeq (S', \mathbf{A}'_{(S',h)})$.

Corollary 1 demonstrates labeling trick's power in the context of local isomorphism. To switch to local $h$-isomorphism, all we need to do is to extract the $h$-hop enclosing subgraph around a node set, and apply labeling trick and GNN only to the extracted subgraph. This is exactly what SEAL does.

## 6 Related work

There is emerging interest in studying the representation power of graph neural networks recently. Xu et al. [35] and Morris et al. [40] first show that the discriminating power of GNNs performing neighbor aggregation is bounded by the 1-WL test. Many works have since been proposed to increase the power of GNNs by simulating higher-order WL tests [40–42]. However, most previous works focus on improving GNN's whole-graph representation power. Little work has been done to analyze GNN's node/link representation power. Srinivasan and Ribeiro [26] first formally studied the difference between structural representations of nodes and links. Although showing that structural node representations of GNNs cannot perform link prediction, their way to learn structural link representations is to give up GNNs and instead use Monte Carlo samples of node embeddings learned by network embedding methods. In this paper, we show that GNNs combined with labeling trick can as well learn structural link representations, which reassures using GNNs for link prediction.

Many works have implicitly assumed that if a model can learn node representations well, then combining the pairwise node representations can also lead to good link representations [43, 19, 44]. However, we argue in this paper that simply aggregating node representations fails to discriminate a large number of non-isomorphic links, and with labeling trick the aggregation of structural node representations leads to structural link representations. Li et al. [21] proposed distance encoding (DE), whose implementations based on $S$-discriminating distances can be shown to be specific labeling tricks. They proved that DE can improve 1-WL-GNNs' discriminating power, enabling them to differentiate almost all $(S, \mathbf{A})$ tuples sampled from $r$-regular graphs. Our paper contributes to an important aspect that Li et al. [21] overlooked: 1) Our theory focuses on the gap between a GNN's single-node and multi-node representation power. We show even a GNN has maximum node representation power, it still fails to learn structural representations of node sets unless combined with a labeling trick. However, the theory of DE cannot explain this. 2) Our theory is not restricted to $r$-regular graphs, but applies to any graphs. 3) Our theory points out that a valid labeling trick is not necessarily distance based—it need only be permutation equivariant and $S$-discriminating. More discussion on the difference between DE's theory and the theory in this paper is given in Appendix E.

You et al. [45] also noticed that structural node representations of GNNs cannot capture the dependence (in particular distance) between nodes. To learn position-aware node embeddings, they propose P-GNN, which randomly chooses some anchor nodes and aggregates messages only from the anchor nodes. In P-GNN, nodes with similar distances to the anchor nodes, instead of nodes with similar neighborhoods, have similar embeddings. Thus, P-GNN cannot learn structural node/link representations. P-GNN also cannot scale to large datasets. You et al. [38] later proposed ID-GNN. As discussed in Section 4.3, ID-GNN's node coloring scheme can be seen as a partial labeling trick.

Finally, although labeling trick is formally defined in this paper, various forms of specific labeling tricks have already been used in previous works. To our best knowledge, SEAL [20] proposes the first labeling trick, which is designed to improve GNN's link prediction power. It is later adopted in inductive knowledge graph completion [28] and matrix completion [29], and is generalized into DE [21] which works for $|S| > 2$ cases. Wan et al. [46] use labeling trick for hyperedge prediction.

## 7 Experiments

In this section, we use a two-node task, link prediction, to empirically validate the effectiveness of labeling trick for multi-node representation learning. We use four link existence prediction datasets in Open Graph Benchmark (OGB) [27]: `ogbl-ppa`, `ogbl-collab`, `ogbl-ddi`, and `ogbl-citation2`. These datasets are open-sourced, large-scale (up to 2.9M nodes and 30.6M edges), adopt realistic train/validation/test splits, and have standard evaluation procedures, thus providing an ideal place to benchmark an algorithm's realistic link prediction power. The evaluation metrics include Hits@$K$ and MRR. Hits@$K$ counts the ratio of positive edges ranked at the K-th place or above against all the negative edges. MRR (Mean Reciprocal Rank) computes the reciprocal rank of the true target node against 1,000 negative candidates, averaged over all the true source nodes. Both metrics are

Table 1: Results for `ogbl-ppa`, `ogbl-collab`, `ogbl-ddi` and `ogbl-citation2`.

| Category | Method | ogbl-ppa Hits@100 (%) | | ogbl-collab Hits@50 (%) | | ogbl-ddi Hits@20 (%) | | ogbl-citation2 MRR (%) | |
|---|---|---|---|---|---|---|---|---|---|
| | | Validation | Test | Validation | Test | Validation | Test | Validation | Test |
| Non-GNN | **CN** | 28.23±0.00 | 27.6±0.00 | 60.36±0.00 | 61.37±0.00 | 9.47±0.00 | 17.73±0.00 | 51.19±0.00 | 51.47±0.00 |
| | **AA** | 32.68±0.00 | 32.45±0.00 | 63.49±0.00 | 64.17±0.00 | 9.66±0.00 | 18.61±0.00 | 51.67±0.00 | 51.89±0.00 |
| | **MLP** | 0.46±0.00 | 0.46±0.00 | 24.02±1.45 | 19.27±1.29 | – | – | 29.03±0.17 | 29.06±0.16 |
| | **Node2vec** | 22.53±0.88 | 22.26±0.88 | 57.03±0.52 | 48.88±0.54 | 32.92±1.21 | 23.26±2.09 | 61.24±0.11 | 61.41±0.11 |
| | **MF** | 32.28±4.28 | 32.29±0.94 | 48.96±0.29 | 38.86±0.29 | 33.70±2.64 | 13.68±4.75 | 51.81±4.36 | 51.86±4.43 |
| Plain GAE | **GraphSAGE** | 17.24±2.64 | 16.55±2.40 | 56.88±0.77 | 54.63±1.12 | 62.62±0.37 | 53.90±4.74 | 82.63±0.23 | 82.60±0.36 |
| | **GCN** | 18.45±1.40 | 18.67±1.32 | 52.63±1.15 | 47.14±1.45 | 55.50±2.08 | 37.07±5.07 | 84.79±0.23 | 84.74±0.21 |
| | **GCN+LRGA** | 25.75±2.82 | 26.12±2.35 | 60.88±0.59 | 52.21±0.72 | 66.75±0.58 | 62.30±9.12 | 66.48±1.61 | 66.49±1.59 |
| Labeling Trick | **GCN+DE** | 36.31±3.59 | 36.48±3.78 | 64.13±0.16 | 64.44±0.29 | 29.85±2.25 | 26.63±6.82 | 60.17±0.63 | 60.30±0.61 |
| | **GCN+DRNL** | 46.43±3.03 | 45.24±3.95 | 64.51±0.42 | 64.40±0.45 | 29.47±1.54 | 22.81±4.93 | 81.07±0.30 | 81.27±0.31 |
| | **SEAL** | **51.25**±2.52 | **48.80**±3.16 | **64.95**±0.43 | **64.74**±0.43 | 28.49±2.69 | 30.56±3.86 | **87.57**±0.31 | **87.67**±0.32 |

higher the better. We include more details and statistics of these datasets in Appendix F. Our code is available at `https://github.com/facebookresearch/SEAL_OGB`.

**Baselines.** We use the following baselines for comparison. We use 5 non-GNN methods: CN (common neighbor), AA (Adamic-Adar), MLP, MF (matrix factorization) and Node2vec. Among them, CN and AA are two simple link prediction heuristics based on counting common neighbors, which are used for sanity checking. We use 3 plain GAE baselines: GraphSAGE [44], GCN [19], and GCN+LRGA [47]. These methods use the Hadamard product of pairwise node representations output by a GNN as link representations, without using a labeling trick. Finally, we compare 3 GNN methods using labeling tricks: GCN+DE [21], GCN+DRNL, and SEAL [20]. GCN+DE/GCN+DRNL enhance GCN with the DE/DRNL labeling trick. SEAL uses a GCN and the DRNL labeling trick, with an additional subgraph-level readout SortPooling [9]. More details are in Appendix G. Moreover, we test the zero-one labeling trick in our ablation experiments. Results can be found in Appendix H.

**Results and discussion.** We present the main results in Table 1. Firstly, we can see that GAE methods without labeling trick do not always outperform non-GNN methods. For example, on `ogbl-ppa` and `ogbl-collab`, simple heuristics CN and AA outperform plain GAE methods by large margins. This suggests that GAE methods cannot even learn simple neighborhood-overlap-based heuristics, verifying our argument in Introduction. In contrast, when GNNs are enhanced by labeling trick, they are able to beat heuristics. With labeling trick, GNN methods achieve new state-of-the-art performance on 3 out of 4 datasets. In particular, we observe that SEAL outperforms GAE and positional embedding methods, sometimes by surprisingly large margins. For example, in the challenging `ogbl-ppa` graph, SEAL achieves an Hits@100 of 48.80, which is **87%-195% higher** than GAE methods without using labeling trick. On `ogbl-ppa`, `ogbl-collab` and `ogbl-citation2`, labeling trick methods also achieve state-of-the-art results.

Despite obtaining the best results on three datasets, we observe that labeling trick methods do not perform well on `ogbl-ddi`. `ogbl-ddi` is considerably denser than the other graphs. It has 4,267 nodes and 1,334,889 edges, resulting in an average node degree of 500.5. In `ogbl-ddi`, labeling trick methods fall behind GAE methods using trainable node embeddings. One possible explanation is that `ogbl-ddi` is so dense that a practical GNN with *limited expressive power* is hard to inductively learn any meaningful structural patterns. In comparison, the transductive way of learning free-parameter node embeddings makes GAEs no longer focus on learning inductive structural patterns, but focus on learning node embeddings. The added parameters also greatly increase GAEs' model capacity. An interesting future topic is to study how to improve labeling tricks' performance on dense graphs. Appendix H presents more ablation experiments to study the power of different labeling tricks, the effect of subgraph pooling, and the number of hops/layers.

## 8  Conclusions

In this paper, we proposed a theory of using GNNs for multi-node representation learning. We first pointed out the key limitation of a common practice in previous works that directly aggregates node representations as a node set representation. To address the problem, we proposed labeling trick which gives target nodes distinct labels in a permutation equivariant way. We proved that labeling trick enables GNNs to learn most expressive structural representations of node sets, which formally characterizes GNNs' maximum multi-node representation learning ability. Experiments on four OGB datasets verified labeling trick's effectiveness.

## Acknowledgements

The authors greatly thank the actionable suggestions from the reviewers. Li is partly supported by the 2021 JP Morgan Faculty Award and the National Science Foundation (NSF) award HDR-2117997.

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
