To prove the first direction, we notice that with an injective AGG,

$$\text{AGG}(\{\text{GNN}(i, \mathbf{A}^{(S)}))|i \in S\}) = \text{AGG}(\{\text{GNN}(i, \mathbf{A}'^{(S')}))|i \in S'\})$$

$$\implies \exists\, v_1 \in S, v_2 \in S', \text{ such that } \text{GNN}(v_1, \mathbf{A}^{(S)}) = \text{GNN}(v_2, \mathbf{A}'^{(S')}) \tag{1}$$

$$\implies (v_1, \mathbf{A}^{(S)}) \simeq (v_2, \mathbf{A}'^{(S')}) \quad \text{(because GNN is node-most-expressive)} \tag{2}$$

$$\implies \exists\, \pi \in \Pi_n, \text{ such that } v_1 = \pi(v_2), \mathbf{A}^{(S)} = \pi(\mathbf{A}'^{(S')}). \tag{3}$$

Remember $\mathbf{A}^{(S)}$ is constructed by stacking $\mathbf{A}$ and $\mathbf{L}^{(S)}$ in the third dimension, where $\mathbf{L}^{(S)}$ is a tensor satisfying: $\forall \pi \in \Pi_n$, (1) $\mathbf{L}^{(S)} = \pi(\mathbf{L}^{(S')}) \Rightarrow S = \pi(S')$, and (2) $S = \pi(S'), \mathbf{A} = \pi(\mathbf{A}') \Rightarrow \mathbf{L}^{(S)} = \pi(\mathbf{L}^{(S')})$. With $\mathbf{A}^{(S)} = \pi(\mathbf{A}'^{(S')})$, we have both

$$\mathbf{A} = \pi(\mathbf{A}'), \ \mathbf{L}^{(S)} = \pi(\mathbf{L}^{(S')}).$$

Because $\mathbf{L}^{(S)} = \pi(\mathbf{L}^{(S')}) \Rightarrow S = \pi(S')$, continuing from Equation (3), we have

$$\text{AGG}(\{\text{GNN}(i, \mathbf{A}^{(S)})|i \in S\}) = \text{AGG}(\{\text{GNN}(i, \mathbf{A}'^{(S')})|i \in S'\})$$

$$\implies \exists\, \pi \in \Pi_n, \text{ such that } \mathbf{A} = \pi(\mathbf{A}'), \ \mathbf{L}^{(S)} = \pi(\mathbf{L}^{(S')}) \tag{4}$$

$$\implies \exists\, \pi \in \Pi_n, \text{ such that } \mathbf{A} = \pi(\mathbf{A}'), \ S = \pi(S') \tag{5}$$

$$\implies (S, \mathbf{A}) \simeq (S', \mathbf{A}'). \tag{6}$$

Now we prove the second direction. Because $S = \pi(S'), \mathbf{A} = \pi(\mathbf{A}') \Rightarrow \mathbf{L}^{(S)} = \pi(\mathbf{L}^{(S')})$, we have:

$$(S, \mathbf{A}) \simeq (S', \mathbf{A}')$$

$$\implies \exists\, \pi \in \Pi_n, \text{ such that } S = \pi(S'), \mathbf{A} = \pi(\mathbf{A}') \tag{7}$$

$$\implies \exists\, \pi \in \Pi_n, \text{ such that } S = \pi(S'), \mathbf{A} = \pi(\mathbf{A}'), \mathbf{L}^{(S)} = \pi(\mathbf{L}^{(S')}) \tag{8}$$

$$\implies \exists\, \pi \in \Pi_n, \text{ such that } S = \pi(S'), \mathbf{A}^{(S)} = \pi(\mathbf{A}'^{(S')}) \tag{9}$$

$$\implies \exists\, \pi \in \Pi_n, \text{ such that } \forall v_1 \in S, v_2 \in S', v_1 = \pi(v_2), \text{ we have } \text{GNN}(v_1, \mathbf{A}^{(S)}) = \text{GNN}(v_2, \mathbf{A}'^{(S')}) \tag{10}$$

$$\implies \text{AGG}(\{\text{GNN}(v_1, \mathbf{A}^{(S)})|v_1 \in S\}) = \text{AGG}(\{\text{GNN}(v_2, \mathbf{A}'^{(S')})|v_2 \in S'\}), \tag{11}$$

which concludes the proof.

$\square$

# B   Proof of Theorem 2

We restate Theorem 2: In any non-attributed graph with $n$ nodes, if the degree of each node in the graph is between 1 and $\mathcal{O}(\log^{\frac{1-\epsilon}{2h}} n)$ for any constant $\epsilon > 0$, then there exists $\omega(n^{2\epsilon})$ many pairs of non-isomorphic links $(u, w), (v, w)$ such that an $h$-layer 1-WL-GNN gives $u, v$ the same representation, while with labeling trick the 1-WL-GNN gives $u, v$ different representations.

*Proof.* Our proof has two steps. First, we would like to show that there are $\omega(n^\epsilon)$ nodes that are locally $h$-isomorphic (see Definition 8) to each other. Then, we prove that among these nodes,

there are at least $\omega(n^{2\epsilon})$ pairs of nodes such that there exists another node constructing locally $h$ non-isomorphic links with either of the two nodes in each node pair.

**Step 1.** Consider an arbitrary node $v$ and denote the subgraph induced by the nodes that are at most $h$-hop away from $v$ as $G_v^{(h)}$ (the $h$-hop enclosing subgraph of $v$). As each node is with degree $d = \mathcal{O}(\log^{\frac{1-\epsilon}{2h}} n)$, then the number of nodes in $G_v^{(h)}$, denoted by $|V(G_v^{(h)})|$, satisfies

$$|V(G_v^{(h)})| \leq \sum_{i=0}^{h} d^i = \mathcal{O}(d^h) = \mathcal{O}(\log^{\frac{1-\epsilon}{2}} n).$$

We set the max $K = \max_{v \in V} |V(G_v^{(h)})|$ and thus $K = \mathcal{O}(\log^{\frac{1-\epsilon}{2}} n)$.

Now we expand subgraphs $G_v^{(h)}$ to $\bar{G}_v^{(h)}$ by adding $K - |V(G_v^{(h)})|$ independent nodes for each node $v \in V$. Then, all $\bar{G}_v^{(h)}$ have the same number of nodes, which is $K$, though they may not be connected graphs.

Next, we consider the number of non-isomorphic graphs over $K$ nodes. Actually, the number of non-isomorphic graph structures over $K$ nodes is bounded by $2^{\binom{K}{2}} = \exp(\mathcal{O}(\log^{1-\epsilon} n)) = o(n^{1-\epsilon})$.

Therefore, due to the pigeonhole principle, there exist $n/o(n^{1-\epsilon}) = \omega(n^\epsilon)$ many nodes $v$ whose $\bar{G}_v^{(h)}$ are isomorphic to each other. Denote the set of these nodes as $V_{iso}$, which consist of nodes that are all locally $h$-isomorphic to each other. Next, we focus on looking for other nodes to form locally $h$-non-isomorphic links with nodes $V_{iso}$.

**Step 2.** Let us partition $V_{iso} = \cup_{i=1}^{q} V_i$ so that for all nodes in $V_i$, they share the same first-hop neighbor sets. Then, consider any pair of nodes $u, v$ such that $u, v$ are from different $V_i$'s. Since $u, v$ share identical $h$-hop neighborhood structures, an $h$-layer 1-WL-GNN will give them the same representation. Then, we may pick one $u$'s first-hop neighbor $w$ that is not $v$'s first-hop neighbor. We know such $w$ exists because of the definition of $V_i$. As $w$ is $u$'s first-hop neighbor and is not $v$'s first-hop neighbor, $(u, w)$ and $(v, w)$ are not isomorphic. With labeling trick, the $h$-layer 1-WL-GNN will give $u, v$ different representations immediately after the first message passing round due to $w$'s distinct label. Therefore, we know such a $(u, w), (v, w)$ pair is exactly what we want.

Based on the partition $V_{iso}$, we know the number of such non-isomorphic link pairs $(u, w)$ and $(v, w)$ is at least:

$$Y \geq \prod_{i,j=1, i \neq j}^{q} |V_i||V_j| = \frac{1}{2}\left[ (\sum_{i=1}^{q} |V_i|)^2 - \sum_{i=1}^{q} |V_i|^2 \right]. \tag{12}$$

Because of the definitions of the partition, $\sum_{i=1}^{q} |V_i| = |V_{iso}| = \omega(n^\epsilon)$ and the size of each $V_i$ satisfies

$$1 \leq |V_i| \leq d_w = \mathcal{O}(\log^{\frac{1-\epsilon}{2h}} n),$$

where $w$ is one of the common first-hop neighbors shared by all nodes in $V_i$ and $d_w$ is its degree.

By plugging in the range of $|V_i|$, Eq.12 leads to

$$Y \geq \frac{1}{2}(\omega(n^{2\epsilon}) - \omega(n^\epsilon)\mathcal{O}(\log^{\frac{1-\epsilon}{2h}} n)) = \omega(n^{2\epsilon}),$$

which concludes the proof.

$\square$

## C  Proof of Lemma 1

We restate Lemma 1: Given a graph with $n$ nodes, a 1-WL-GNN takes up to $\mathcal{O}(n)$ message passing layers to discriminate all the nodes that 1-WL can discriminate.

*Proof.* We first note that after one message passing layer, 1-WL-GNN gives different embeddings to any two nodes that 1-WL gives different colors to after one iteration. So we only need to show how many iterations 1-WL takes to converge in any graph.

Note that if two nodes are given different colors by 1-WL at some iteration (they are discriminated by 1-WL), their colors are always different in any future iteration. And if at some iteration, all nodes' colors are the same as their colors in the last iteration, then 1-WL will stop (1-WL fails to discriminate any more nodes and has converged). Therefore, before termination, 1-WL will increase its total number of colors by at least 1 after every iteration. Because there are at most $n$ different final colors given an $n$-node graph, 1-WL takes at most $n - 1 = \mathcal{O}(n)$ iterations before assigning all nodes different colors.

Now it suffices to show that there exists an $n$-node graph that 1-WL takes $\mathcal{O}(n)$ iterations to converge. Suppose there is a path of $n$ nodes. Then by simple calculation, it takes $\lceil n/2 \rceil$ iterations for 1-WL to converge, which concludes the proof. □

## D    Comparisons between DRNL and DE

In this section, we discuss the relationships and differences between DRNL [20] and DE [21] (using shortest path distance). Although they are theoretically equivalent in the context of link prediction, there are some subtle differences that might result in significant performance differences.

Suppose $x$ and $y$ are the two end nodes of the link. **DRNL** (Double Radius Node Labeling) always assigns label 1 to $x$ and $y$. Then, for any node $i$ with $(d(i, x), d(i, y)) = (1, 1)$, it assigns a label 2. Nodes with radius $(1, 2)$ or $(2, 1)$ get label 3. Nodes with radius $(1, 3)$ or $(3, 1)$ get 4. Nodes with $(2, 2)$ get 5. Nodes with $(1, 4)$ or $(4, 1)$ get 6. Nodes with $(2, 3)$ or $(3, 2)$ get 7. So on and so forth. In other words, DRNL iteratively assigns larger labels to nodes with a larger radius w.r.t. both the two end nodes. The DRNL label $f_l(i)$ of a node $i$ can be calculated by the following hashing function:

$$f_l(i) = 1 + \min(d_x, d_y) + (d/2)[(d/2) + (d\%2) - 1], \tag{13}$$

where $d_x := d(i, x)$, $d_y := d(i, y)$, $d := d_x + d_y$, $(d/2)$ and $(d\%2)$ are the integer quotient and remainder of $d$ divided by 2, respectively. This hashing function allows fast closed-form computations of DRNL labels. For nodes with $d(i, x) = \infty$ or $d(i, y) = \infty$, DRNL assigns them a null label 0. Later, the one-hot encoding of these labels are fed to a GNN as the initial node features, or equivalently, we can feed the raw integer labels to an embedding layer first.

Instead of encoding $(d(i, x), d(i, y))$ into a single integer label, **DE** (distance encoding) directly uses the vector $[d(i, x), d(i, y)]$ as a size-2 label for node $i$. Then, these size-2 labels will be transformed to two-hot encoding vectors to be used as the input node features to GNN. Equivalently, we can also input the size-2 labels to an embedding layer and use the sum-pooled embedding as the initial node features.

These two ways of encoding $(d(i, x), d(i, y))$ theoretically have the same expressive power. However, DRNL and DE have some subtle differences in their implementations. The **first difference** is that DE sets a maximum distance $d_{\max}$ (a small integer such as 3) for each $d(i, x)$ or $d(i, y)$, i.e., if $d(i, x) \geq d_{\max}$, DE will let $d(i, x) = d_{\max}$. This potentially can avoid some overfitting by reducing the number of possible DE labels as claimed in the original paper [21].

The **second difference** is that when computing the distance $d(i, x)$, DRNL will temporarily mask node $y$ and all its edges, and when computing the distance $d(i, y)$, DRNL will temporarily mask node $x$ and all its edges. The reason for this "masking trick" is because DRNL aims to use the pure distance between $i$ and $x$ without the influence of $y$. If we do not mask $y$, $d(i, x)$ will be upper bounded by $d(i, y) + d(x, y)$, which obscures the "true distance" between $i$ and $x$ and might hurt the node labels' ability to discriminate structurally-different nodes. As we will show in Appendix H, this masking trick has a great influence on the performance, which explains DE's inferior performance than DRNL in our experiments.

As we will show in Table 1, DRNL has significantly better performance than DE on some datasets. To study what is the root cause for these in-principle equivalent methods's different practical performance, we propose **DE$^+$**, which adopts DRNL's masking trick in DE. We also try to not set a maximum distance in DE$^+$. This way, there are no more differences in terms of the expressive power between DE$^+$ and DRNL. And we indeed observed that DE$^+$ is able to catch up with DRNL in those datasets where DE does not perform well, as we will show in Appendix H.2.

# E   More discussion on the differences between DE's theory and ours

Inspired by the empirical success of SEAL [20], Li et al. [21] proposed distance encoding (DE). It generalizes SEAL's distance-based node labeling (DRNL) for link prediction to arbitrary node set prediction, and theoretically studies how the distance information improves 1-WL-GNN's discriminating power. The main theorem in [21] (Theorem 3.3) proves that under mild conditions, a 1-WL-GNN combined with DE can discriminate any $(S, \boldsymbol{A}), (S', \boldsymbol{A'})$ pair sampled uniformly from all $r$-regular graphs, with high probability. This is a significant result, as 1-WL-GNN's discriminating power is bounded by 1-WL, which fails to discriminate any nodes or node sets from $r$-regular graphs. DE's theory shows that with DE we can break the limit of 1-WL and 1-WL-GNN on this major class of graphs where without DE they always fail.

Despite the success, DE's theory also has several limitations. Firstly, its analysis focuses on the space of random graphs (in particular regular graphs that 1-WL-GNNs fail to represent well). Secondly, DE's theory does not answer whether a GNN combined with DE can learn structural representations, which are the core for joint node set prediction tasks such as link prediction according to [26]. Thirdly, although DE's definition (Definition 3.1 of [21]) only requires permutation invariance, its theory and practical implementations require distance-based node labeling. It is unknown whether other node labeling tricks (including those do not rely on distance) are also useful.

Our theory partly addresses these limitations and is orthogonal to DE's theory, as: 1) We define labeling trick, which is not necessarily distance-based. We show a valid labeling trick need only be permutation equivariant and target-node-set-discriminating. 2) We show with a sufficiently expressive GNN, labeling trick enables learning structural representations of node sets, answering the open question in [26] which DE's theory fails to address. 3) We show labeling trick's usefulness for arbitrary graphs, instead of only regular graphs.

Nevertheless, we are uncertain whether DE's power for regular graphs can transfer to any valid labeling trick, including those not based on distance. Thus, we leave an open question here for future research: whether an arbitrary labeling trick enables a 1-WL-GNN to discriminate any $(S, \boldsymbol{A}), (S', \boldsymbol{A'})$ pair sampled uniformly from all $r$-regular graphs, with high probability? Our guess is that the answer is yes for $|S| = 1$ and $|S| = 2$. This is because, with an injective message passing layer, we can propagate the unique labels of $S$ to other nodes, thus "recovering" the distance information through iterative message passing. We leave a rigorous proof or disproof to future work.

# F   More details about the datasets

We compare the link prediction performance of different baselines on `ogbl-ppa`, `ogbl-collab`, `ogbl-ddi`, and `ogbl-citation2`. Among them, `ogbl-ppa` is a protein-protein association graph where the task is to predict biologically meaningful associations between proteins. `ogbl-collab` is an author collaboration graph, where the task is to predict future collaborations. `ogbl-ddi` is a drug-drug interaction network, where each edge represents an interaction between drugs which indicates the joint effect of taking the two drugs together is considerably different from their independent effects. `ogbl-citation2` is a paper citation network, where the task is to predict missing citations. We present the statistics of these OGB datasets in Table 2. More information about these datasets can be found in [27].

Table 2: Statistics and evaluation metrics of OGB link prediction datasets.

| Dataset | #Nodes | #Edges | Avg. node deg. | Density | Split ratio | Metric |
|---|---|---|---|---|---|---|
| `ogbl-ppa` | 576,289 | 30,326,273 | 73.7 | 0.018% | 70/20/10 | Hits@100 |
| `ogbl-collab` | 235,868 | 1,285,465 | 8.2 | 0.0046% | 92/4/4 | Hits@50 |
| `ogbl-ddi` | 4,267 | 1,334,889 | 500.5 | 14.67% | 80/10/10 | Hits@20 |
| `ogbl-citation2` | 2,927,963 | 30,561,187 | 20.7 | 0.00036% | 98/1/1 | MRR |

We choose OGB datasets for benchmarking our methods because these datasets adopt realistic train/validation/test splitting methods, such as by resource cost in laboratory (`ogbl-ppa`), by time (`ogbl-collab` and `ogbl-citation2`), and by drug target in the body (`ogbl-ddi`). They are also large-scale (up to 2.9M nodes and 30.6M edges), open-sourced, and have standard evaluation metrics.

OGB has an official leaderboard[4], too, providing a place to fairly compare different methods' link prediction performance.

## G    More details about the baselines

We include baselines achieving top places on the OGB leaderboard. All the baselines have their open-sourced code and paper available from the leaderboard. We adopt the numbers published on the leaderboard if available, otherwise we run the method ourselves using the open-sourced code. Note that there are other potentially strong baselines that we have to omit here, because they cannot easily scale to OGB datasets. For example, we have contacted the authors of P-GNN [45], and confirmed that P-GNN is not likely to scale to OGB datasets due to the computation of all-pairs shortest path distances.

All the compared methods are in the following. We briefly describe how each method obtains its final node representations.

- **MLP**: Node features are directly used as the node representations without considering graph structure.
- **Node2vec** [30, 43]: The node representations are the concatenation of node features and Node2vec embeddings.
- **MF** (Matrix Factorization): Use free-parameter node embeddings trained end-to-end as the node representations.
- **GraphSAGE** [44]: A GAE method with GraphSAGE as the GNN.
- **GCN** [19]: A GAE method with GCN as the GNN.
- **LRGA** [47]: A GAE method with LRGA-module-enhanced GCN.
- **GCN+DE**: Apply GCN to the DE [21] labeled graphs.
- **GCN+DRNL**: Apply GCN to the DRNL [20] labeled graphs.
- **SEAL** [20]: The same as GCN+DRNL with an additional subgraph-level readout. Note that we reimplemented SEAL in this paper with a greatly improved efficiency and flexibility than the original implementation[5]. The code will be released in the future.

Except SEAL, all models use the Hadamard product between pairwise node representations as the link representations. The link representations are fed to an MLP for final prediction. All the GAE methods' GNNs have 3 message passing layers with 256 hidden dimensions, with a tuned dropout ratio in $\{0, 0.5\}$. All the labeling trick methods (GCN+DE, GCN+DRNL and SEAL) extract 1-hop enclosing subgraphs. The GCNs in GCN+DRNL and GCN+DE also use 3 message passing layers with 256 hidden dimensions for consistency. The GNN in SEAL follows the DGCNN in the original paper, which has 3 GCN layers with 32 hidden dimensions each, plus a SortPooling layer [9] and several 1D convolution layers after the GCN layers to readout the subgraph. The use of a subgraph-level readout instead of only reading out two nodes is not an issue for SEAL, because 1) the two center nodes' information is still included in the output of the subgraph-level readout, and 2) the inclusion of additional neighborhood node representations may help learn better neighborhood features than only reading out two center nodes. As we will show in Appendix H.3, a subgraph-level readout sometimes improves the performance.

The `ogbl-ddi` graph contains no node features, so MLP is omitted, and the GAE methods here use free-parameter node embeddings as the GNN input node features and train them together with the GNN parameters. For labeling trick methods, the node labels are input to an embedding layer and then concatenated with the node features (if any) as the GNN input. Note that the original SEAL can also include pretrained node embeddings as additional features. But according to [26], node embeddings bring no additional value given structural representations. This is also consistent with our observation and the experimental results of [20], where including node embeddings gives no better results. Thus, we give up node embeddings in SEAL.

For the baseline GCN+LRGA, its default hyperparameters result in out of GPU memory on `ogbl-citation2`, even we use an NVIDIA V100 GPU with 32GB memory. Thus, we have to reduce its hidden dimension to 16 and matrix rank to 10. It is possible that it can achieve better

---

[4]`https://ogb.stanford.edu/docs/leader_linkprop/`
[5]`https://github.com/muhanzhang/SEAL`

performance with a larger hidden dimension and larger matrix rank using a GPU with a larger memory.

We implemented the labeling trick methods (GCN+DE, GCN+DRNL and SEAL) using the PyTorch Geometric [48] package. For all datasets, labeling trick methods only used a fixed 1% to 10% of all the available training edges as the positive training links, and sampled an equal number of negative training links randomly. Labeling trick methods showed excellent performance even without using the full training data, which indicates its strong inductive learning ability. Due to using different labeled subgraphs for different links, labeling trick methods generally take longer running time than GAE methods. On the largest `ogbl-citation2` graph, SEAL takes about 7 hours to finishing its training of 10 epochs, and takes another 28 hours to evaluate the validation and test MRR each. For `ogbl-ppa`, SEAL takes about 20 hours to train for 20 epochs and takes about 4 hours for evaluation. The other two datasets are finished within hours.

## H    Ablation study

In this section, we conduct several ablation experiments to more thoroughly study the effect of different components around labeling trick on the final link prediction performance.

### H.1    How powerful is the zero-one labeling trick?

Firstly, we aim to understand how powerful the proposed zero-one labeling (Definition 6) is. Although zero-one labeling is a also valid labeling trick that theoretically enables a node-most-expressive GNN to learn structural representations, in practice our GNNs may not be expressive enough. Then how will the zero-one labeling trick perform compared to those more sophisticated ones such as DE and DRNL? We conduct experiments on `ogbl-collab` and `ogbl-citation2` to answer this question. In Table 3, we compare GCN (1-hop) + all-zero labeling (not a valid labeling trick), GCN (1-hop) + zero-one labeling trick, and GCN (1-hop) + DRNL. All methods use the same 3 GCN layers with 256 hidden dimensions, 1-hop enclosing subgraphs, and Hadamard product of the two end node representations as the link representations. All the remaining settings follow those of GCN+DRNL.

Table 3: Ablation study on the power of the zero-one labeling trick.

| Method | ogbl-collab Hits@50 (%) | | ogbl-citation2 MRR (%) | |
|---|---|---|---|---|
| | Validation | Test | Validation | Test |
| **GCN (1-hop) + all-zero labeling** | 24.35±1.28 | 25.92±1.47 | 36.97±0.56 | 36.98±0.57 |
| **GCN (1-hop) + zero-one labeling trick** | 44.45±1.39 | 44.79±1.26 | 38.73±0.86 | 38.78±0.88 |
| **GCN (1-hop) + DRNL** | 64.51±0.42 | 64.40±0.45 | 81.07±0.30 | 81.27±0.31 |
| **GIN (1-hop) + zero-one labeling trick** | 60.31±0.81 | 59.48±1.17 | 78.32±1.07 | 78.50±1.08 |

From Table 3, we can see that GCN+zero-one labeling trick indeed has better performance than GCN without labeling trick, which aligns with our theoretical results that even a simple zero-one labeling is also a valid labeling trick that enables learning structural representations. Nevertheless, the zero-one labeling trick is indeed less powerful than DRNL, as shown by the performance gaps especially on the `ogbl-citation2` dataset. We are then interested in figuring out what could cause such large performance differences between two (both valid) labeling tricks, because as Theorem 1 shows, any valid labeling trick can enable a node-most-expressive GNN to learn structural link representations.

We suspect that the insufficient expressive power of GCN is the cause. Therefore, we change GCN to Graph Isomorphism Network (GIN) [35]. By replacing the linear feature transformations in GCN with MLPs, GIN is one of the most expressive GNNs based on message passing. The results are shown in the last column of Table 3. As we can see, GIN (1-hop) + zero-one labeling trick has much better performance than GCN (1-hop) + zero-one labeling trick, and is almost catching up with GCN (1-hop) + DRNL. The results very well align with our theory—as long as we have a sufficiently expressive GNN, even a simple zero-one labeling trick can be very powerful in terms of enabling learning structural representations. Nevertheless, in practice when we only have less powerful GNNs, we should better choose those more sophisticated labeling tricks such as DE and DRNL for better link prediction performance.

## H.2 DE vs. DE$^+$ vs. DRNL

In Appendix D, we have discussed the differences of the implementations of DE and DRNL. That is, although DE and DRNL are equivalent in theory, there are two differences in their implementations: 1) DE sets a maximum distance $d_{\max}$ (by default 3) while DRNL does not, and 2) DRNL masks the other end node when computing the distances to one end node and vice versa, while DE does not. To study whether it is these implementation differences between DE and DRNL that result in the large performance differences in Table 1, we propose **DE$^+$** which no longer sets a maximum distance in DE and additionally does the masking trick like DRNL. We compare DE, DE$^+$, and DRNL on `ogbl-ppa` and `ogbl-citation2` (where DE shows significantly lower performance than DRNL in Table 1). All of them use GCN as the GNN with the same hyperparameters. The results are shown in Table 4.

Table 4: Comparison of DE, DE$^+$ and DRNL.

| | `ogbl-ppa` Hits@100 (%) | | `ogbl-citation2` MRR (%) | |
| --- | --- | --- | --- | --- |
| **Method** | Validation | **Test** | Validation | **Test** |
| **GCN+DE** ($d_{\max} = 3$) | 36.31±3.59 | 36.48±3.78 | 60.17±0.63 | 60.30±0.61 |
| **CCN+DE$^+$** ($d_{\max} = 3$) | 47.17±1.84 | 45.70±3.46 | 74.75±1.18 | 75.00±1.20 |
| **CCN+DE$^+$** ($d_{\max} = \infty$) | 45.81±3.53 | 43.88±5.18 | 79.37±4.50 | 78.85±0.17 |
| **GCN+DRNL** | 46.43±3.03 | 45.24±3.95 | 81.07±0.30 | 81.27±0.31 |

We can observe that DE$^+$ outperforms DE by large margins. This indicates that the masking trick used in DRNL is very important. Intuitively, temporarily masking the target node $y$ when computing distances to the source node $x$ can give more diverse node labels. Without the masking, $d(i, x)$ will be upper bounded by $d(i, y) + d(x, y)$. Because the distance between $x$ and $y$ can be small in positive links, without the masking $d(i, x)$ will be restricted to small numbers, which hurts their ability to detect subtle differences between nodes' relative positions within the subgraph. Nevertheless, the benefit of the masking trick is not observed in smaller datasets such as `ogbl-collab` (Table 1).

We can also find that DE$^+$ without setting a maximum distance has very close performance to DRNL, which aligns with our discussion in Appendix D. By removing the maximum distance restriction, DE$^+$ essentially becomes DRNL. However, there are still small performance differences, possibly because DRNL has a larger embedding table than DE$^+$ (DRNL's maximum label is larger) which results in a slightly larger model capacity. Nevertheless, this can be alleviated by doubling the embedding dimension of DE$^+$. In summary, we can conclude that the masking trick used in DRNL is crucial to the performance on some datasets. Compared to DE, DE$^+$ and DRNL show better practical performance. Studying more powerful labeling tricks is also an important future direction.

## H.3 Is a subgraph-level readout useful?

In Table 1, we observe that SEAL is generally better than GCN+DRNL. SEAL also uses GCN and the DRNL labeling trick, so the main difference is the subgraph-level readout in SEAL. That is, instead of only reading out the two center nodes' representations as the link representation, SEAL performs a readout over all the nodes in the enclosing subgraph. Here we study this effect further by testing whether a subgraph-level sum-pooling readout is also useful. We replace the Hadamard product of two center node representations in GCN+DRNL with the sum over all node representations within the enclosing subgraph. The results are shown in Table 5.

Table 5: Ablation study on subgraph-level readout.

| | `ogbl-collab` Hits@50 (%) | | `ogbl-citation2` MRR (%) | |
| --- | --- | --- | --- | --- |
| **Method** | Validation | **Test** | Validation | **Test** |
| **GCN+DRNL** | 64.51±0.42 | 64.40±0.45 | 81.07±0.30 | 81.27±0.31 |
| **GCN+DRNL (sum-pooling)** | 64.64±0.24 | 63.26±0.35 | 84.98±0.23 | 85.20±0.26 |
| **SEAL** | 64.95±0.43 | 64.74±0.43 | 87.57±0.31 | 87.67±0.32 |

As we can see, using sum-pooling has a similar effect to the SortPooling in SEAL, i.e., it greatly improves the performance on `ogbl-citation2`, while slightly reduces the performance on `ogbl-collab`. This means, using a subgraph-level readout can sometimes be very helpful. Although according to Theorem 1 we only need to aggregate the representations of the two center nodes (two end nodes of the link) as the link representation, in practice, because our GNNs only have limited expressive power, reading out all nodes within the enclosing subgraph could help GNNs learn better subgraph-level features thus better detecting the target link's local $h$-isomorphism class. Such subgraph representations can be more expressive than only the two center nodes' representations, especially when the number of message passing layers is small so that the center nodes have not gained enough information from the whole subgraph.

## H.4   Is it helpful to make number of layers larger than number of hops?

In all labeling trick methods, we have used a fixed enclosing subgraph hop number $h = 1$, and a fixed number of message passing layers $l = 3$. Using a number of message passing layers larger than the number of hops is different from the practice of previous work. For example, in GAE, we always select $h = l$ hops of neighbors if we decide to use $l$ message passing layers. So is it really helpful to use $l > h$? Intuitively, using $l > h$ layers can make GNNs more sufficiently absorb the entire enclosing subgraph information and learn better link representations. Theoretically, as we have shown in Lemma 1, to reach the maximum representation power of 1-WL-GNN, we need to use $\mathcal{O}(n)$ number of message passing layers, where $n$ is the number of nodes in the enclosing subgraph. Thus, using $l > h$ can enhance GNN's representation power and learn more expressive link representations.

Table 6: Ablation study on subgraph-level readout.

| Method | ogbl-ppa Hits@100 (%) | | ogbl-collab Hits@50 (%) | | ogbl-citation2 MRR (%) | |
|---|---|---|---|---|---|---|
| | Validation | **Test** | Validation | **Test** | Validation | **Test** |
| **GCN+DRNL** ($l = 3$) | 46.43±3.03 | 45.24±3.95 | 64.51±0.42 | 64.40±0.45 | 81.07±0.30 | 81.27±0.31 |
| **GCN+DRNL** ($l = 1$) | 31.59±2.79 | 33.57±3.06 | 64.38±0.13 | 63.95±0.42 | 77.77±0.42 | 78.02±0.44 |

To validate the above , we conduct experiments on GCN+DRNL by using $l = 1$ message passing layers (and still $h = 1$). The results are shown in Table 6. As we can observe, using $l = 1$ results in lower performance than using $l = 3$ in all three datasets. On `ogbl-collab`, this effect is very small. However, on `ogbl-ppa` and `ogbl-citation2`, the performance gaps are significant. These results demonstrate the usefulness of using more message passing layers than hops.

Nevertheless, we are unsure whether it is still helpful to make $l > h$ when we use a large $h$, such as $h = 2$ or $h = 3$. We cannot generally verify this because increasing $h$ will exponentially increase our subgraph sizes. And considering the huge computation cost on two relatively large datasets `ogbl-ppa` and `ogbl-citation2`, using $h = 1$ is currently the maximum $h$ we can afford. We thus only conduct experiments using different $h$'s on the smallest `ogbl-collab` dataset. We have tried different combinations of $(l, h)$ from $(1, 1)$ all the way up to $(4, 3)$, and the testing scores are consistently around 63 to 64. This seems to indicate increasing $h$ or $l$ is not helpful in this dataset. Nevertheless, `ogbl-collab` may not be representative enough to derive a general conclusion. For example, in the original SEAL paper [20], the authors found using $h = 2$ is helpful for many datasets. Thus, fully answering this question might need further investigations. But when $h = 1$, we can conclude that using $l > h$ is better.