# OpenReview forum: "Labeling Trick: A Theory of Using Graph Neural Networks for Multi-Node Representation Learning"
_NeurIPS.cc/2021/Conference — NeurIPS 2021 Poster_

### Official Review · Reviewer_KGAE · 2021-07-14

**Rating:** 6
**Confidence:** 3

**Summary:**

This paper introduces a new concept "labeling trick" along with multiple ways of using or decorating it such as "valid labeling trick" and "zero-one labeling trick". It proves that popular link prediction algorithms such as SEAL, Distance Encoding, and ID-GNN can all be considered as specific implementations of the labeling trick. So the new concept, though was not explicitly referred in the literature, has demonstrated its significance. The paper offers a theoretical analysis to explain the outperformance of the labeling trick over plain GAE. The authors' response addressed most of my concerns when I had a difficulty to digest the new concepts and combine them with the theory. An important suggestion is offering a figure to illustrate the labeling trick and show the specifications of the three algorithms (how they are sharing the labeling trick philosophy and still having their uniqueness of implementation). That will significantly help broad readers catch the point of this paper at the beginning.

**Ethical Concerns:**

I don't have any concerns.

**Limitations And Societal Impact:**

Not included or discussed.

**Main Review:**

Originality: In the authors' response, they clarified the difference between the labeling trick and graph data augmentation. It's good to know the authors will discuss this point in the paper revision. Quoting from the authors' response:
"The labeling tensor L(S) is not “new data added into the original data A”. The labeled tensor “A concat L(S)” is the new data, and this new data is not “added” into the original data. Instead, it replaces the original data A."
I suggest the authors to clarify the difference between "concat+replace" and "add" - why concatenating labeling tensor and then replacing is not adding the labeling tensor into the original data A.
Also, I need to point out that the augmentation techniques were narrowly considered (within the computer vision studies) in the authors' responses. In the two papers I've given as well as Lecture on "Graph Augmentation" in Prof. Jure Leskovic's Machine Learning on Graphs course, graph augmentation is looking for the optimal computational graph - some techniques could be adding/dropping node attributes and/or edges beyond flipping or rotating operations in computer vision.
[1] Data augmentation for graph neural networks. AAAI'21.
[2] Nodeaug: Semi-supervised node classification with data augmentation. KDD'20.

Quality: "Structural link representation" must be well defined. The definition is missing. It looks like this concept must include at least "count of common neighbors", but what else? What can GAE learn - and if the things it learned is not about structural links, why it performs better than random guess. The authors' response was not able to address my confusion at this point, though it's not a big case.

Clarity: The authors' response has made the idea and contributions of this work much more clear. I appreciate that. I would still suggest several ways to improve the clarify as aforementioned.

Significance: I encourage exploration on the theory of graph neural networks. It is impactful to generalize the advantage of several algorithms as a new concept and provide theoretical analysis on its advantage over GAE.

I changed my score from 5 to 6.

**Time Spent Reviewing:**

2 hours

---

> ### Author Response · Authors · 2021-08-10
> **Author Response**
>
> We thank the reviewer for pointing out some potential clarity issues and understanding challenges of our paper for GNN practitioners. We answer the related concerns as follows, and will improve on these aspects in the revision.
>
> - Re Q1: “a theoretical analysis is expected to offer comprehensive explanations to observations from existing practical studies. So the concepts should be the ones that have been widely used in the existing studies. It was hard to understand what is "labeling trick" (Def 4) and what is "zero-one labeling trick" (Def 5) which might never be mentioned in any existing paper.”
>
> We respectfully do not fully agree with that.
>
> Firstly, we think the goal of theoretical research need not be only to provide explanations for existing observations; instead, theory can also precede practice, guide practice, and predict new observations.
>
> Secondly, we hold a different opinion on "the concepts used in a theory must have been widely used in existing studies". In many cases, it is necessary to invent some new concepts for 1) rigorously defining something that is a prerequisite of a theorem, 2) summarizing existing concepts into some high-level concept which is more general, and 3) giving an example to better deliver the insight. Both the “labeling trick” and “zero-one labeling trick” are essential concepts in the development of our theory, which cannot be avoided.
>
> Thirdly, our theoretical results not only bring out new theoretical understanding of GNNs, but also provide an explanation to the better performance of SEAL compared to GAE, which actually meets the reviewer’s criterion of “offer explanations to observations”. Moreover, although the “labeling trick” and “zero-one labeling trick” are new concepts, they are essentially generalizing the concepts of “distance encoding” and “Double Radius Node Labeling” in SEAL and related papers. Labeling trick unifies various existing successful techniques for enhancing GNN into a single framework, and insightfully reveals the most basic conditions these techniques have to satisfy in order to achieve the better structural representation ability. Zero-one labeling trick, on the other hand, proposes a simplest form that satisfy the conditions of labeling trick, which serves as an example to illustrate the idea. Thus, both of them are useful, important and necessary.
>
> - Re Q2: “By concatenating A and L(S), we get the new labeled graph ..." This is exactly the idea of data augmentation: Given the original graph A, create some new data from A, and then add the new data into the original data. I suggest the authors to review the literature of graph (data) augmentation methods for graph learning models, especially those for graph neural networks”
>
> We thank the reviewer for suggesting the graph data augmentation line of research. Although labeling trick looks like data augmentation by altering node/edge attributes, it has important differences from data augmentation as follows: 1) Data augmentation increases the amount of data by adding slightly modified copies of the training data or newly created synthetic data from existing data, while labeling trick directly replaces the original data with the altered (labeled) ones without increasing the amount of training data. 2) Data augmentation is often done by stochastic techniques like random cropping, flipping, erasing and noise injection, while labeling trick is a deterministic method which returns a labeled graph deterministically for each given graph. 3) Data augmentation acts as a regularizer to help reduce overfitting, so that the machine learning model is robust to slight changes of the input data. In comparison, labeling trick completely gives up the original data during training --- both of its training and testing are done on the labeled graphs. Its purpose is not regularization, but a conditioning method to increase the link representation power of GNNs.
>
> The reviewer describes labeling trick as “Given the original graph A, create some new data from A, and then add the new data into the original data.” This might suggest the source of the misunderstanding. The labeling tensor L(S) is not “new data added into the original data A”. The labeled tensor “A concat L(S)” is the new data, and this new data is not “added” into the original data. Instead, it replaces the original data A. We hope the above can clear things up. We will discuss these differences between labeling trick and data augmentation in the revised paper.
>
> - Re Q3: “I am not convinced that "GAE cannot learn structural link representations". If it could not capture the structural information/patterns, it would not deliver a significantly better performance than random guess. The example in Figure 1 can only tell the GAE's structural representation learning algorithm is not perfect.”
>
> Please check our definition of “structural link representation” in Definition 2 and its explanation (line 126 - 130). Structural link representation is not equivalent to “representations that capture some structural information/patterns”. Instead, it requires to preserve all the structural information up to link isomorphism. The structural representations of two links are the same if and only if these two links are isomorphic. In other words, structural link representation has to be “perfect”. The example in Figure 1 tells that GAE’s structural representation learning is not perfect, which essentially shows that GAE cannot learn structural link representations.
>
> - Re Q4: “Then this paper moved on to discuss the labeling trick of SEAL. It's not clear how to perform a perfect structural representation learning from graphs - in other words, fully utilizing the graph information in node/link representations.”
>
> Our Theorem 1 exactly discusses how to perform a perfect structural representation learning from graphs. By learning over a labeled graph (through labeling trick), a node-most-expressive GNN is able to learn structural link representations, allowing to perfectly map isomorphic links to the same representation and distinguish between non-isomorphic links.
>
> - Re Q5: “There was no explanation on why only link prediction was discussed but not node-level or graph-level tasks.”
>
> This paper focuses on exploring GNN’s representation power for higher-order node sets, such as links, hyperedges, etc. We have clearly stated the motivation in the abstract and introduction.
>
> - Re Q6: “Clarity: The paper is a bit hard to follow, as it is based on quite a few new jargons. There must be a way to re-organize the paper to make it easier to understand by practitioners (who are using GNNs to solve real problems).”
>
>  We are sorry that the reviewer feels a bit hard to follow our paper. We did not define the new concepts to intentionally make the paper hard to follow. Instead, these concepts are necessary in order to develop our theory in a rigorous and general way. Besides, we have tried our best to explain each concept or theorem in plain words immediately after its formal definition. We will keep improving the clarity of the paper and making the core take-aways more salient in the paper.
>
> - Re Q7: “Significance: I encourage exploration on the theory of graph neural networks. However, as the paper is hard to follow, unfortunately, I learn very little from it.”
>
> We thank the reviewer for encouraging theoretical work on GNNs as well as honestly acknowledging the difficulty in following the paper. We will try our best to make it easier-to-read in the revision.

---

### Official Review · Reviewer_b9cR · 2021-07-16

**Rating:** 9
**Confidence:** 4

**Summary:**

This paper dives into the study of some theoretical properties of GNNs and their training for the task of link prediction. In particular, it addresses the expressive power of GNNs for learning structural link representations, and points out that even with full node representation power, GNNs can fail to represent edges adequately. The authors show that the graph autoencoder method for link prediction cannot learn structural link predictions, resulting in some non-isomorphic graph edges getting the same representation. This is due to the fact that GAE methods learn node representations independently, as opposed to conditioning the representation of the target nodes on each other. The authors continue to describe a labeling trick that helps GNNs learn structural link representations, show that the trick works, and further prove that the SEAL method uses a valid labeling trick. The authors conduct experiments on three datasets to verify their theoretical results, and compare their method with a class of very basic link prediction approaches (MLP, node2vec,...) and another class of popular GNN approaches (GCN, GraphSAGE, SEAL,...). The experiments confirm their findings.


**Ethical Concerns:**

No ethical concerns.

**Limitations And Societal Impact:**

Limitations address adequately across the paper/appendix.


**Main Review:**

This is a very good paper that addresses important theoretical questions about GNNs that go beyond WL testing.
The paper is very well written. The notation is consistent and laid out with care. Definitions are clear and formal. The authors motivate the reasons for studying their problem with examples, which are very helpful in following the buildup of the lemmas leading to the main conclusions in the paper. In the Related Work section, the authors cover the range of existing work on the topic and place their contribution very nicely among existing work and lay out its differences clearly.
Experiments are well described and adequate. The authors provide many  details in the appendix, and show results of various ablations that explore questions about number of layers vs number of hops, subgraph-level readout, power of zero-one labeling, ...etc.

The paper enhances our core understanding of limitations of some GNN methods for link prediction. It also provides a labeling technique that unifies some of the existing attempts to improve the expressive ability of graph link representation. The theoretical discussions are presented general enough to apply to generalized link prediction (e.g. in hypergraphs).


Minor comments:
- Line 210: "rest nodes" should be something like "remaining nodes"
- Line 378: "then" should be "them"


**Time Spent Reviewing:**

3

---

> ### Author Response · Authors · 2021-08-10
> **Author Response**
>
> We thank the reviewer for the very insightful review which thoroughly summarizes our paper and contributions. We will address the typos and keep improving our paper in the revision!

---

### Official Review · Reviewer_gL7o · 2021-07-17

**Rating:** 7
**Confidence:** 4

**Summary:**

This paper studies the problem of using GNN for link prediction. The contributions are mainly two folds. First, it studies the limitation of using GAE for link prediction, which directly aggregate a pair of nodes embedding. Secondly, it proposes the labeling trick which is sufficiently expressive and unifies some of the existing frameworks. The proposed method is general for high-order node set prediction where link prediction is a special case.

**Limitations And Societal Impact:**

I think one major limitation is the computational cost of the proposed labeling framework. In comparison to GAE, where node representation for all nodes are learned first and link prediction could be made for multiple links at the same time, the labeling trick need to compute node representations under a given node $S$. For predicting multiple links, that is computationally cost. Would this be implemented in parallel for link prediction for multiple node pairs?

**Main Review:**

This paper first points out the limitation of GAE for link prediction, showing that GAE learn same representations for isomorphism nodes, this would give non-isomorphism links the same link representation. To address that, this paper proposes a zero-one labeling trick that labels the nodes in the target set $S$. This proposed trick unifies some existing GNN methods for link prediction. The expressive power of the labeling trick is also studied with theoretical guaranteed. I think the proposed method is general enough as it unifies the existing methods such as SEAL and could be applied for high-order link set prediction or hypergraph prediction. My mainly concern for the proposed method, which I also mentioned in the limitation section, is that the labeling trick need to perform node embeddings for each given target set $S$. For GAE, we could first obtain node representations for all nodes and then make link predictions for any pair of nodes. While the labeling trick seems to be link-specific, and should be computationally cost when applying to large scale data and predicting multiple links. I wonder in your experiments, how does the order of computational cost for GAE-based method and labeling based method?

**Time Spent Reviewing:**

5

---

> ### Author Response · Authors · 2021-08-10
> **Author Response**
>
> We thank the reviewer for their insightful review. About the computational cost of labeling trick, yes, it indeed requires link-specific GNN computation and node embeddings. This results in a higher computational complexity than GAEs as a price for the better link representation ability. We have discussed this complexity in Appendix F, and reported the empirical computation time (in hours) in Appendix H. Overall labeling trick methods can take up to 35 hours to finish the training and inference on the largest dataset, while GAE methods only take hours. One of our future work is to reduce the computation cost of labeling tricks. On the other hand, for very large graphs that cannot fit into the GPU memory, even GAE methods have to use mini-batch training to compute node embeddings for each $S$ separately. Then GAE will have a similar cost to labeling trick.
>
> In terms of parallelization, yes, we implemented labeling tricks to be able to predict multiple links in parallel (through feeding multiple enclosing subgraphs in a batch). Please refer to the submitted code for details.

---

### Official Review · Reviewer_4QvJ · 2021-07-17

**Rating:** 6
**Confidence:** 4

**Summary:**

This paper focuses on the link prediction task of graph neural networks. The authors delve into the structural link representation and propose a labeling trick that enables existing GAE models to distinguish between non-isomorphic links.

**Limitations And Societal Impact:**

Yes.

**Main Review:**

The authors note the performance gap between GAE and SEAL on the link prediction task, and argue that the former can only learn the structural node representation, while the latter can further learn the structural link representation, which is significant for the link prediction problem. Based on the above analysis, the authors put forward the labeling trick, which theoretically proved to be a good solution to this problem. This strategy can be seamlessly integrated with existing GAE approaches so that they can learn structural link representation to distinguish between non-isomorphic links in the graph. The authors also demonstrate that SEAL, ID-GNN, and other methods are essentially special cases of the labeling trick. In general,  the paper examines a very interesting problem and provides a useful but not novel enough solution.

**Time Spent Reviewing:**

an hour

---

> ### Author Response · Authors · 2021-08-10
> **Author Response**
>
> We thank the reviewer for their positive comment and nice summarization of our paper! We will keep improving it.

---

### Decision · Program_Chairs · 2021-09-27

**Decision:**

Accept (Poster)

**Comment:**

The paper focuses on GNNs for link prediction and sheds light on the performance gap between GAE and SEAL by identifying a key limitation in GAE. To overcome this limitation, a labeling trick is proposed that enables the unification of existing frameworks and that generalizes to high-order node set prediction.

The AC and reviewers carefully examined the author feedback and all agree that this paper makes some very pertinent contributions leading to a better understanding of GNNs for link prediction.

It would be important to incorporate into the main text the remark on computational complexity from the author response. As noted by the authors, the labeling trick can result in significant computational overhead, and such a limitation should not be completely relegated to the supplements.